# Biomolecular Screening of *Pimpinella anisum* L. for Antioxidant and Anticholinesterase Activity in Mice Brain

**DOI:** 10.3390/molecules28052217

**Published:** 2023-02-27

**Authors:** Aamir Mushtaq, Fatima Habib, Rosana Manea, Rukhsana Anwar, Umar Farooq Gohar, Muhammad Zia-Ul-Haq, Mobasher Ahmad, Claudia Mihaela Gavris, Liana Chicea

**Affiliations:** 1Department of Pharmaceutical Sciences, Government College University, Lahore 54000, Pakistan; 2Gulab Devi Institute of Pharmacy, Gulab Devi Educational Complex, Lahore 54000, Pakistan; 3Faculty of Medicine, Transilvania University of Brasov, 500036 Brasov, Romania; 4Department of Pharmacology, Punjab University College of Pharmacy, University of the Punjab, Lahore 54000, Pakistan; 5Institute of Industrial Biotechnology, Government College University, Lahore 54000, Pakistan; 6Office of Research, Innovation and Commercialization, Lahore College for Women University, Lahore 54000, Pakistan; 7Faculty of Medicine, University “Lucian Blaga” Sibiu, 550169 Sibiu, Romania

**Keywords:** dementia, brain, apiol, cholinergic, natural products, memory

## Abstract

Hundreds of the plants have been explored and evaluated for antioxidant and anti-amnesic activities, so far. This study was designed to report the biomolecules of *Pimpinella anisum* L. for the said activities. The aqueous extract of dried *P. anisum* seeds was fractionated via column chromatography and the fractions so obtained were assessed for the inhibition of acetylcholinesterase (AChE) via in vitro analysis. The fraction which best inhibited AChE was so named as the *P. anisum* active fraction (P.aAF). The P.aAF was then chemically analyzed via GCMS, which indicated that oxadiazole compounds were present in it. The P.aAF was then administered to albino mice to conduct the in vivo (behavioral and biochemical) studies. The results of the behavioral studies indicated the significant (*p* < 0.001) increase in inflexion ratio, by the number of hole-pokings through holes and time spent in a dark area by P.aAF treated mice. Biochemical studies demonstrated that the oxadiazole present in P.aAF on one hand presented a noteworthy reduction in MDA and the AChE level and on the other hand promoted the levels of CAT, SOD and GSH in mice brain. The LD_50_ for P.aAF was calculated as 95 mg/Kg/p.o. The findings thus supported that the antioxidant and anticholinesterase activities of *P. anisum* are due to its oxadiazole compounds.

## 1. Introduction

Alzheimer’s disease (AD) is one of the major and distinct forms of dementia in the geriatric population. The incidence of this disease is 1/8 in American people above the age of 60 years. Its prevalence rate may double in next 20 years [1]. The deficiency of cholinergic neurons in specific region of brain (hippocampus) is the principal hallmark of this disease. The pathophysiologic factor of AD is acetylcholine (Ach) [2]. Oxidative stress such as that caused by reactive oxygen species also contributes to progression of disease [3]. The unavailability of antioxidants in the routine diet is the major cause of a high level of oxidative stress which may lead to dementia. Similarly, acetylcholinesterase (AChE) is a very important enzyme which contributes to pathogenesis in AD by reducing Ach levels in the brain [4].The excessive breakdown of Ach at synapsis can be prevented by the inhibition of AChE [5]. Tacrine and rivastigmine (inhibitors of acetylcholinesterase) are widely practiced all over the world to overcome this cholinergic dysfunction. However, their less significant effects on memory performance as well as adverse effects have recommended the need for some alternate therapies for AD [6]. Natural products have been progressively investigated for the isolation of therapeutic substances for the last few decades. The safe and toxicity-free profile of natural herbs always encourage the use of crude natural products for neurodegenerative brain disorders [7]. Herbs are the richest source of anti-inflammatory and anti-oxidant substances which may be used for the prevention of dementia along with rejuvenation of brain and other body parts [8]. Many essential oils, volatile oils, aromatic extracts and herbal products have been scientifically investigated for anti-inflammatory and antioxidant properties and they have been proven very beneficial for brain health [9]. Most of the relevant scientific work has reported that progressive memory loss is due to cholinergic crisis and plant extracts have been of great interest to scientists to overcome this problem. In a study conducted by Torre et al. (2022), 90 extracts from 30 native plants of Spain were scientifically tested, out of which 21 extracts showed very high anticholinesterase activity. The phenolic contents of the extracts were reported to possess anti AChE and antioxidant potential [10]. Similarly, ginseng and licorice are famous at treating cerebral inflammation in patients with AD or other cognitive disorders [11].

*P. anisum* (Umbelliferae) is an annual herb and widely cultivated in Asia Minor and India [12]. Most commonly, it is used for the treatment of abdominal cramps, constipation, colic, duodenal ulcers, dysmenorrhea, nausea, inflammation [13] and epilepsy [14]. Eugenol, α-terpineol, 1,8-cineol, α-pinene [15], limonene, trans anethole [16], palmitic acid, linoleic acid and oleic acid have been reported in dried *P. anisum* seeds, which possess strong anti-cholinesterase activity.

It has already been reported in previous studies that *P. anisum* possesses strong anti-epileptic, cerebroprotective [14] and enzyme-inhibition potential [17], which is why we selected this plant and found its anti-amnesic potential [18]. This study was designed to report the biomolecules present in the aqueous extract of dried *P. anisum* seeds which are responsible for the memory-enhancing potential. We also made attempts to report the acute toxicity of the plant.

## 2. Results

### 2.1. Frasctionation by Column Chromatography

Aqueous extract of dried *P. anisum* seeds was fractionated using column chromatography and various solvents in different ratios. We obtained forty fractions which were identified using the thin-layer chromatographic (TLC) technique. We mixed all the fractions and purified them by repassing them through the column, and finally thirteen fractions were obtained (Figure 1), upon which in vitro testing was conducted.

### 2.2. In Vitro Testing of Purified Fractions

It was observed that only fraction number F-8 exhibited in vitro anticholinesterase activity and all other remaining fractions fail to do so (Table 1). Fraction F-8 was selected for in vivo studies and was named as the *P. anisum* active fraction (P.aAF). The chemical constituents present in P.aAF were analyzed via GC-MS analysis.

### 2.3. Results of GC-MS Analysis

The GC-MS analysis of the *P. anisum* active fraction (P.aAF) indicated the presence of oxadiazole compounds, i.e., 1,2,5 oxadiazole, 1-benzylbenzimidazole 3-oxide, apiol and cyclohexanone (Table 2). The chromatogram obtained through GC-MS analysis is shown in Figure 2.

### 2.4. Findings of Behavioral Studies

The findings of behavioral studies through different paradigms indicated that animals treated with the *P. anisum* active fraction (P.aAF) exhibited a significantly (*p* < 0.001) higher inflexion ratio in comparison to the scopolamine-treated hyper-amnesic mice. The light/dark test paradigm indicated that active-fraction-treated animals explored the dark compartment and remained most of the time in dark area in comparison to amnesic control mice. Similarly, it was observed via the hole board test that active-fraction-treated mice presented a significantly (*p* < 0.001) higher number of pokings through the holes of the apparatus as compared to scopolamine-induced hyper-amnesic mice. The detailed numerical values of behavioral studies are recorded in Table 3.

### 2.5. Findings of Biochemical Studies

The biochemical investigations indicated that the brain homogenates of active-fraction-treated mice presented a significant (*p* < 0.001) reduction in MDA and AChE content in comparison to scopolamine-treated mice. Similarly, a significant (*p* < 0.001) improvement in the level of SOD, CAT and GSH was observed in active-fraction-treated mice (Table 4). However, the level of ChAT remained unchanged in active-fraction-treated mice (Table 5).

### 2.6. Acute Toxicity

The findings of the acute toxicity study indicated that no deaths were recorded up to a dose of 100 mg/Kg/p.o. However, the animals started dying at a dose of 150 mg/Kg/p.o. (3 out of 5 animals died in this group). All animals died in Group-4 by administration of a single acute toxic dose of 200 mg/Kg/p.o. From this data the LD_50_ for the *P. anisum* active fraction (P.aAF) was calculated as 95 mg/Kg/p.o.

## 3. Discussion

The current study was performed to report the active ingredients of *P. anisum* for the prevention of dementia. Initially, methanolic and aqueous extracts of *P. anisum* were tested for behavioral and biochemical studies. This manuscript is a continuation of this series which involved the identification of the active moieties responsible for anti-cholinesterase and anti-oxidant activities in mice brains. The GC-MS analysis indicated that the *P. anisum* active fraction (P.aAF) contained apiol, 1-benzylbenzimidazole 3-oxid, cyclohexanone and a heterocyclic aromatic compound, 1,2,5 oxadiazole (Table 2). 1,2,5 oxadiazole belongs to the azole family and possesses antioxidant, anti-epileptic, anti-diabetic, anti-microbial, anti-Alzheimer’s and enzyme-inhibition activities [19].

The findings of biochemical studies indicated that the *P. anisum* active fraction (P.aAF) at a dose of 7 mg/Kg/p.o. produced a significant (*p <* 0.001) increase in the level of SOD, CAT and GSH in mice brains, while the MDA contents were reduced significantly. This indicates the strong antioxidant potential of the P.aAF 7 mg/Kg/p.o. dose. Previous studies have explained that oxadiazole compounds prevent the oxidative load of the body by scavenging free radicals. Hence, they can be used as potent antioxidant substances [20]. Studies have also proved that oxadiazole compounds prevent the cellular damage of the body by reducing lipid peroxidation reactions. Oxygen free radicals produced in the body participate in lipid peroxidation reactions and, as a result, the MDA contents are increased in the brain and cerebrospinal fluid. Thus, MDA can be used as a valuable marker to investigate the oxidizing load of the body [21]. It was observed that the MDA contents were significantly high in scopolamine-treated mice. Scopolamine was used to induce amnesia in mice and is actually responsible for the increase in the oxidizing load in mice brains. The overproduction of hydroxyl free radicals results in the propagation of lipid peroxidation reactions and, as a consequence, the MDA contents are raised in brain homogenates [22]. It is clear from the biochemical investigation that P.aAF significantly (*p <* 0.001) lowered the MDA contents of mice brains by the inhibition of lipid peroxidation reactions. The significant reduction in the level of MDA contents (Table 4) by administration of P.aAF might be due the presence of the oxadiazole present in it. Similarly, the reference drug piracetam also protected neurons from oxidative stress. It involves the improvement in mitochondrial function along with enhancement of ATP production. It also enhances the membrane fluidity of the hippocampus and prevents the neuron from oxidative stress [23].

Similarly, it was also observed that the animals treated only with scopolamine presented a marked reduction in the levels of SOD, GSH and CAT. These are important endogenous anti-oxidants which prevent the body from free radicals and reactive substances. A reduction in the level of natural antioxidants promoted free-radical-induced apoptosis in mice brains. Similarly, the administration of scopolamine in mice resulted in the over-expression of cytokines such as interleukin-1β and other inflammatory proteins, which induced amnesia via the cholinergic dysfunction of the mice brain [24]. The findings of biochemical studies suggested that the pretreatment of animals with P.aAF prevented the oxidizing load of the mice brain by increasing GSH, CAT and SOD level. The reduced GSH is responsible for the donation of its electrons to the reactive oxygen species, thus neutralizing them so they do not produce damage. Similarly, it also prevents lipid peroxidation reactions and minimizes the damage produced by heavy metals. On the other hand, SOD is responsible for neutralizations of the superoxide species, providing a first-line defense against oxidizing species. Increased levels of CAT in mice brains reduces hydrogen peroxide into molecular oxygen and simple water and, hence, protects the brain from damage [25,26].

Our findings also suggested that P.aAF possesses a strong anticholinesterase activity in brain homogenates of the mice (Table 4). This enzymatic inhibition might be attributed to the oxadiazole compounds of *P. anisum*. Acetylcholinesterase inhibition by oxadiazole compounds is responsible for providing the plant with its memory-enhancing potential. Previous studies indicated that 1,2,4 oxadiazole is much more famous for its broad range of pharmacological potentials [19], and various derivatives of oxadiazole compounds have been synthesized by substitution at certain positions of the oxadiazole ring [27]. Substitution at the R^1^ and R^2^ side chains on 1,2,4 oxadiazoles have imparted them with a strong anticholinesterase activity and 1,3,4 oxadiazoles have been proven to be parent compounds which offer both acetylcholinesterase- and butyrylcholinesterase-inhibition activities [28].

The behavioral studies supported the biochemical mechanism of memory enhancement of the test substance. The results of elevated plus maze (EPM) paradigm indicated that the inflexion ratio was significantly (*p* < 0.001) improved in the P.aAF-treated group compared to the amnesic control group. The increase in inflexion ratio is the hallmark of improvement in memory. The EPM paradigm is one of the widely employed paradigms to assess the memory-enhancement effect of natural products [29]. Similarly, the reduction in contact time in a light compartment and increased hole-poking by the P.aAF-treated mice indicated the improvement in retention power, as given in Table 3.

Acute toxicity studies were performed via the administration of a single acute toxic dose of the *P. anisum* active fraction (P.aAF) to different groups of mice, and the LD_50_ was calculated as 95 mg/Kg/p.o. for P.aAF. The therapeutic index was calculated as 13.57 and it was observed that the administration of P.aAF in a toxic dose (150 mg/Kg/p.o.) produced marked behavioral changes in animals. Spasticity of muscles and hypersecretions of saliva (Table 6) were observed in these animals. This might be due to the presence of apiol in the *P. anisum* active fraction. It has been reported in previous studies that apiol is a very potent appetite stimulant and also promotes the enzymatic secretions of the digestive tract [30]. From all the above discussions, it is quite clear that the GC-MS analysis reported the presence of oxadiazole compounds in the *P. anisum* active fraction (P.aAF). These oxadiazole compounds not only possess a strong anti-oxidant potential but are also responsible for the anticholinesterase activity of the plant. Thus, the memory-enhancing effect of the *P. anisum* plant is attributed to the presence of its oxadiazole compounds, which inhibit acetyl cholinesterase on one hand, and promote the level of natural anti-oxidants in the mice brain on the other hand.

## 4. Materials and Methods

### 4.1. Extraction and Fractionation via Column Chromatography

#### 4.1.1. Botanical Material

The dried seeds of *P. anisum* were purchased from a grocery shop and they were identified by a botanist at GC-University Lahore. The specimen was assigned an identification code (Herb.Bot.3385) and was preserved in the herbarium of GC-University Lahore.

#### 4.1.2. Extraction and Fractionation

The dried seeds were ground into a coarse-sized powder and the methanolic extract was prepared using the maceration technique [31]. The methanolic extract was used as a mother fraction of the plant extract, from which we extracted *n*-hexane, chloroform, ethyl acetate, *n*-butanol and aqueous fractions of the extract. All other fractions were left behind and only the aqueous fraction of *P. anisum* was purified using column chromatography, because our previous study reported that among all fractions, only the aqueous fraction was capable of producing memory-enhancing effects [18].

#### 4.1.3. Column Chromatography

For column chromatography, a medium-sized glass column was used, which was packed with almost 10 g of silica gel-60 after making its slurry in the same solvent as was used for the mobile phase. The column was packed properly and solvent was run through the column following the standard procedures [32]. Then, the known quantity of aqueous extract of *P. anisum* was dissolved in the solvent and its uniform mixture was prepared. It was then loaded into the glass column quite carefully, with the help of pipette, in such a way that a uniform thick layer of the sample mixture was formed above the slurry. Finally, the solvent was added above the sample layer in a sufficient quantity and the stopper of the column was opened to obtain the separated fractions in the flask. Different solvents were used in different ratios to obtain the separation of the contents of the aqueous extract of *P. anisum*. Using this technique, we got 16 fractions that were collected in small glass vials and were labelled properly. They were then used for further studies.

### 4.2. In Vitro Anti-Cholinesterase Activity

The purified fractions obtained through column chromatography were then analyzed for in vitro anticholinesterase activity. We used the NAFB micro-well plate assay technique by making a solution mixture by dissolving 0.25 mg of β-naphthyl acetate in 1 mL of methanol. We took 10 µL of purified plant fraction and mixed it with 50 µL of the above-prepared β-naphthyl acetate solution. The temperature of the reaction mixture was reduced and maintained at 4 °C, and then we added 200 µL of acetylcholine esterase (3.33 U/mL) solution to it and it was incubated for 40 to 50 min in the same conditions. Then, a solution of Fast Blue B salt solution was prepared by dissolving 2.5 mg of it in 1 mL of distilled water. Finally, 10 µL of the Fast Blue B salt was added to the reaction mixture dropwise, and a change in color was observed. If the color of the reaction mixture turned purple, it indicated that there was no inhibition of the cholinesterase enzyme by the test substance, while no change in color specified a strong anticholinesterase action of the test substance [33].

### 4.3. GC-MS Analysis

The purified fraction which produced a strong in vitro anticholinesterase activity was named as the *P. anisum* active fraction (P.aAF), which was selected and analyzed using the GC-MS technique. We used GC-MS equipment (Agilent 6890N) with the following specifications: capillary column (TR-5-MS), 30 Mts dimensions, 0.25 µm film thickness, and 0.25 mm internal diameter. The carrier gas used in the GC-MS was 99.99% helium and the flow speed of the mobile was adjusted to 1 mL/min. The temperature of the oven was raised from 40 to 250 °C at a rate of 10 °C/min and the temperature of the ion source was fixed at 200 °C, the injector at 250 °C and the detector at 280 °C at the time of the sample injection. The test sample (purified fraction in concentration of 1 mg/mL) was dissolved in methanol and was injected in an aliquot of 2 µL at the already adjusted temperature. The compounds present in the sample were then identified by their molecular masses via interpretation of the GC-MS library, and the structures of compounds were also expressed [31].

### 4.4. Animals

The in vivo studies were performed on Swiss, male albino mice having a weight of 25 ± 5 g. They were housed in polycarbonate cages in the animal house of Punjab University College of Pharmacy (PUCP), Lahore, and in standard living conditions, i.e., a humidity of 50%, a temperature of 25 ± 2 °C and an equal 12 h light and dark span [34]. The ethical approval for the use of animals was granted by the Animal Ethics Committee of PUCP, vide letter no AEC/PUCP/1072 after reviewing the research protocols and study design. The animals were given treatment according to the study design mentioned in Table 7.

### 4.5. In Vivo Behavioral and Biochemical Studies

The purified fraction of *P. anisum* which produced the maximum in vitro anticholinesterase activity (P.aAF) was administered in the animals as per the study design (Table 7) and then behavioral and biochemical studies were conducted. We used an elevated-plus-maze hole-board paradigm and light/dark test apparatuses to perform the behavioral studies. The biochemical analysis was conducted via an estimation of acetylcholinesterase (AChE), malondialdehyde (MDA), catalase (CAT), superoxide dismutase (SOD) and glutathione (GSH) levels in mice brains with all the detailed procedures as mentioned in our previously published manuscripts [31,34].

### 4.6. Choline Acetyltransferase Activity (ChAT)

This part of study was performed on a 2nd set of experimental animals which were treated according to the study design mentioned in Table 8. To perform this test, we prepared the reagent by dissolving 10 µL of 0.5 M sodium phosphate buffer having a pH of 7.2, 7.6 × 10 ^−4^ M neostigmine sulfate solution, 3 M NaCl solution, 1.1 × 10^−3^ Molar EDTA, acetyl coenzyme-A (6.2 × 10^−3^ M prepared in 0.01N HCl) and 1 molar choline chloride. It was then incubated at 37 °C for twenty-five minutes. Then, 100 µL of brain homogenate was mixed thoroughly with 0.2 mL of the reagent and incubated at 37 °C for twenty-five minutes. It was then boiled for two minutes in a water bath and then added up with oxygen less distilled water. The reaction took place and proteins were denatured, which were then separated out via high-speed centrifugation. Finally, 0.5 mL of supernatant was mixed up with 10 µL of 10^−3^ molar 4,4-dithiodipyridine and the absorbance of the mixture was taken at 324 nm using a double-beam UV-visible spectrophotometer [35].

### 4.7. Acute Toxicity Study

We followed OECD guidelines 423, 2001 as mentioned in our previous study [31] for the assessment of the acute toxic effects of the active fraction of *P. anisum* (P.aAF) on female albino mice (25 ± 5 g). Initial pilot studies were performed on mice and we determined the dose range at which death in animals was observed. No death was recorded when the extract was used up to 100 mg/Kg/p.o; however, all the animals died when treated with an acute single dose of 200 mg/Kg/p.o. To find the LD_50,_ 20 animals were equally divided into four groups with *n* = 5. Group-I was kept as normal control while group-II to Group-IV were orally treated with P.aAF in respective doses of 100, 150 and 200 mg/Kg. Animals were kept under observation to record the behavioral and physical changes along with the number of mortalities [36] and, finally, LD_50_ was calculated as [37]: LD_50_ = Least Lethal Dose − Σ (a × b)/*n*.

### 4.8. Statistics

The numerical data were presented as mean ± SEM. One-way ANOVA followed by Dunnett’s test was applied for the multiple comparison and student’s t-test analysis was applied on the data set using Graph Pad Prism software version 7 and a value of *p* of <0.05 was marked as significant.

## 5. Conclusions

Understanding the medicinal importance of dried *P. anisum* seeds, its different extracts were prepared and aqueous extract was purified using column chromatography. The purified fraction of the aqueous extract exhibited marked anticholinesterase and antioxidant activities in albino mice. The chemical analysis indicated the presence of oxadiazole compounds in it. Thus, the aqueous extract of *P. anisum* contains oxadiazole compounds which build up memory by reducing both AChE and oxidizing stress in mice brains. Its LD_50_ was calculated as 95 mg/Kg/p.o and clinical data were limited to ensure its therapeutic safety. Hence, there is a strong need to perform clinical trials to explore the therapeutic potential and safety profile. Studies are further required to investigate the effectiveness of the tested substance for the treatment of other neurological disorders such as Alzheimer’s, along with testing of the toxicity profile in more detail.

## Figures and Tables

**Figure 1 molecules-28-02217-f001:**
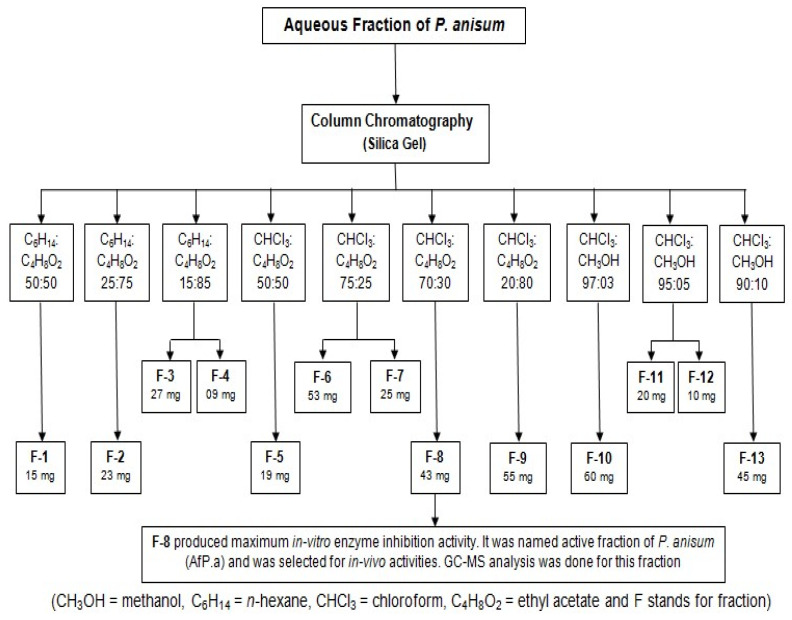
Fractionation of aqueous extract of *P. anisum* into sub fractions via column chromatography.

**Figure 2 molecules-28-02217-f002:**
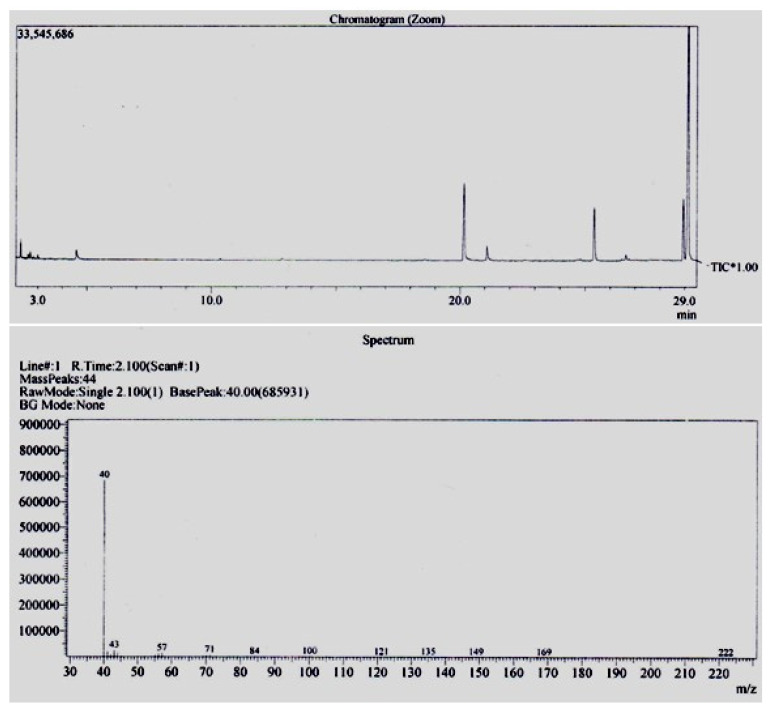
GC-MS analysis of *P. anisum* active fraction (P.aAF).

**Table 1 molecules-28-02217-t001:** In vitro analysis of the purified fraction of *P. anisum* for anti-cholinesterase activity.

Fractions	Color of Solution	AChE Inhibition
F-1	Purple	No
F-2	Purple	No
F-3	Purple	No
F-4	Purple	No
F-5	Purple	No
F-6	Purple	No
F-7	Purple	No
F-8	Colorless	Yes
F-9	Purple	No
F-10	Purple	No
F-11	Purple	No
F-12	Purple	No
F-13	Purple	No

**Table 2 molecules-28-02217-t002:** GC-MS analysis of P.aAF and identification of compounds.

Compound Name	Molecular Formula	Molecular Weight (g/mol)	Mass Peak	Retention Time (min)
1-Benzylbenzimidazole 3-oxide	C_14_H_12_N_2_O	224	43	2.683
Apiol	C_12_H_14_O_4_	222	146	20.158
Cyclohexanone	C_6_H_10_O	98	34	4.567
1,2,5 oxadiazole	C_2_H_2_N_2_O	70	26	2.992

**Table 3 molecules-28-02217-t003:** Effect of P.aAF on enhancement of memory and learning tasks.

	Elevated Plus Maze			Light/Dark Paradigm				Hole-Board	
Group	Day-1	Day-2	I.R	Time Spent on Day-1		Time Spent on Day-2		Day-1	Day-2
	I^1^ (s)	I^2^ (s)		L.Cmpt (s)	D.Cmpt (s)	L.Cmpt (s)	D.Cmpt (s)	n.Pok/5 min	n.Pok/5 min
G-1	23.16 ± 1.17	16.66 ± 0.98	0.26 ± 0.07	50.83 ± 2.42	249.17 ± 2.42	41.50 ± 1.76	258.50 ± 1.76	48.33 ± 1.33	43.33 ± 1.74
G-2	68.11 ± 2.39 ^a^	81.02 ± 2.78 ^a^	ࢤ0.18 ± 0.05 ^a^	173.22 ± 6.99 ^a^	126.78 ± 7.20 ^a^	179.00 ± 5.89 ^a^	121.00 ± 5.88 ^a^	21.00 ± 1.13 ^a^	29.01 ± 1.81 ^a^
G-3	20.83 ± 0.87 ^b,σ^	17.16 ± 1.07 ^b,σ^	0.17 ± 0.04 ^b,σ^	41.66 ± 4.41 ^b,σ^	258.34 ± 4.41 ^b,σ^	31.67 ± 2.47 ^b,σ^	268.33 ± 2.47 ^b,σ^	55.50 ± 2.21 ^b,σ^	45.50 ± 1.80 ^b,σ^
G-4	35.50 ± 0.92 ^b,α^	25.66 ± 1.30 ^b,β^	0.29 ± 0.02 ^b,σ^	62.50 ± 2.14 ^b,σ^	237.50 ± 2.14 ^b,σ^	49.17 ± 2.38 ^b,σ^	250.83 ± 2.38 ^b,σ^	44.00 ± 1.82 ^b,σ^	42.66 ± 1.60 ^b,σ^
G-5	46.16 ± 1.68 ^b,α^	41.16 ± 1.85 ^b,α^	0.10 ± 0.03 ^b,σ^	66.00 ± 5.63 ^b,σ^	234.00 ± 5.63 ^b,σ^	64.00 ± 4.47 ^b,σ^	236.00 ± 4.47 ^b,α^	37.84 ± 2.70 ^b,β^	39.50 ± 1.47 ^b,σ^
G-6	41.00 ± 1.59 ^b,α^	36.16 ± 1.85 ^b,α^	0.11 ± 0.03 ^b,σ^	55.83 ± 6.63 ^b,σ^	244.17 ± 6.63 ^b,σ^	53.33 ± 6.28 ^b,σ^	246.67 ± 6.28 ^b,σ^	39.66 ± 1.76 ^b,γ^	43.33 ± 1.70 ^b,σ^
G-7	36.00 ± 1.59 ^b,α^	31.50 ± 1.92 ^b,α^	0.12 ± 0.04 ^b,σ^	52.50 ± 5.73 ^b,σ^	247.5 ± 5.73 ^b,σ^	51.66 ± 3.80 ^b,σ^	248.34 ± 3.80 ^b,σ^	41.00 ± 1.52 ^b,σ^	44.16 ± 1.24 ^b,σ^

Note: G-1 = Normal Control, G-2 = Amnesic Control, G-3 = Standard Control-A, G-4 = Standard Control-B, G-5 = Experimental Control-I, G-6 = Experimental Control-II, G-7 = Experimental Control-III, I^1^ = Initial Transfer Latency, I^2^ = Retention Transfer Latency, I.R = Inflexion Ratio, L.Cmpt = Light Compartment, D.Cmpt = Dark Compartment and n.Pok = No. of hole-pokings. Data were presented as mean ± SEM (*n* = 6) and statistically analyzed using one-way ANOVA and we used Dunnett’s test for the comparison of the groups. All the groups (G-I and G-3 to G-7) were compared to G-2. The value of *p* ≤ 0.001 was expressed as ^a^ for comparison of G-1 to G-2. The level of significance for comparison of G-3 to G-7 with G-2 was expressed as ^b^, indicating *p* value of ≤0.001, ≤0.01, ≤0.05 and ≥0.05, respectively, and for comparison of G-3 to G-7 with G-1, it was denoted with signs; ^α, β, γ,^ or ^σ^ , indicating *p* value of ≤0.001, ≤0.01, ≤0.05 and ≥0.05, respectively.

**Table 4 molecules-28-02217-t004:** Estimation of levels of AChE, MDA, SOD, CAT and GSH in mice brain.

Groups	AChE μmol/min/mg	MDA nmol/h/g	SOD U/mg of Homogenate	Catalase U/mg of Homogenate	GSH μmol/mg
Group-1	3.79 ± 0.21	1.52 ± 0.11	25.91± 0.61	1.91 ± 0.17	40.21 ± 1.10
Group-2	8.01 ± 0.31 ^a^	6.91 ± 0.40 ^a^	7.61 ± 0.24 ^a^	0.56 ± 0.04 ^a^	17.92 ± 0.33 ^a^
Group-3	3.52 ± 0.30 ^b,σ^	1.40 ± 0.08 ^b,σ^	25.11 ± 0.89 ^b,σ^	2.01 ± 0.06 ^b,σ^	46.99 ± 0.89 ^b,γ^
Group-4	4.61 ± 0.23 ^b,σ^	2.60 ± 0.11 ^b,β^	22.02 ± 0.61 ^b,α^	1.40 ± 0.07 ^b,β^	39.99 ± 1.23 ^b,σ^
Group-5	6.31 ± 0.31 ^b,α^	2.29 ± 0.19 ^b,σ^	20.03 ± 0.61 ^b,α^	1.10 ± 0.07 ^c,α^	37.02 ± 1.89 ^b,σ^
Group-6	5.91 ± 0.19 ^b,α^	1.99 ± 0.13 ^b,σ^	20.21 ± 0.71 ^b,α^	1.30 ± 0.07 ^b,β^	36.99 ± 1.69 ^b,σ^
Group-7	4.40 ± 0.31 ^b,σ^	1.88 ± 0.17 ^b,σ^	21.14 ± 0.49 ^b,α^	1.71 ± 0.06 ^b,σ^	40.07 ± 1.39 ^b,σ^

G-1 = Normal Control, G-2 = Amnesic Control, G-3 = Standard Control-A, G-4 = Standard Control-B, G-5 = Experimental Control-I, G-6 = Experimental Control-II and G-7 = Experimental Control-III. One way-ANOVA followed by Dunnett’s test was applied as a statistical tool to analyze the data and data were expressed as mean ± SEM. After comparison of Group-I to II (denoted by ^a^ for *p* ≤ 0.001), the remaining groups were compared with Group-II (denoted by a for *p* ≤ 0.001). The level of significance was expressed either by ^b^, ^c^, indicating a *p* value of ≤0.001, ≤0.01, ≤0.05 and ≥0.05, respectively. Similarly, Groups III-VII were also compared with Group-I and significance level was expressed either by ^α^, ^β^, ^γ^, or ^σ^, indicating a *p* value of ≤0.001, ≤0.01, ≤0.05 and ≥0.05, respectively.

**Table 5 molecules-28-02217-t005:** Estimation of choline acetyltransferase levels in mice brain.

Groups	Treatment	ChAT (μmol/min/mg)
G-A	Normal Control	12.10 ± 0.89
G-B	Amnesic Control	6.99 ± 0.81 *
G-C	Test Control-A	10.92 ± 0.71 ^ns^
G-D	Test Control-B	10.44 ± 1.31 ^ns^
G-E	Test Control-C	8.81 ± 0.94 ^ns^

Note: All the groups (B, C, D and E) were compared with Group-A. “* represents *p* < 0.001 while ^n.s^ indicates *p* > 0.05”.

**Table 6 molecules-28-02217-t006:** Acute toxicity study showing effect of P.aAf (150 mg/Kg/p.o.) on behavioral and physiological characteristics of mice.

Behavioral Changes	Number of Days													
	I	II	III	IV	V	VI	VII	VIII	IX	X	XI	XII	XIII	XIV
Ataxia	√	√	√	√	√	√	√	√	√	√	√	√	√	√
Strabo Tail	√	√	√	√	√	√	√	√	√	✗	✗	✗	✗	✗
Blanching	√	√	√	√	√	√	√	√	√	√	√	√	√	√
Secretions	√	√	√	√	√	√	√	✗	✗	✗	✗	✗	✗	✗
Convulsions	✗	✗	✗	✗	✗	✗	✗	✗	✗	✗	✗	✗	✗	✗
Salivation	√	√	√	√	√	√	√	✗	✗	✗	✗	✗	✗	✗
Hyperactivity	✗	✗	✗	✗	✗	✗	✗	✗	✗	✗	✗	✗	✗	✗
Rigidity	√	√	√	√	√	✗	✗	✗	✗	✗	✗	✗	✗	✗
Hypnosis	✗	✗	✗	✗	✗	✗	✗	✗	✗	✗	✗	✗	✗	✗
Ptosis	√	√	√	√	√	✗	✗	✗	✗	✗	✗	✗	✗	✗
Irritability	✗	✗	✗	✗	✗	✗	✗	✗	✗	✗	✗	✗	✗	✗
Pilo erection	✗	✗	✗	✗	✗	✗	✗	✗	✗	✗	✗	✗	✗	✗
Muscle Spasm	√	√	√	√	√	√	√	√	√	✗	✗	✗	✗	✗
Loss of Traction	√	√	√	√	√	√	√	√	√	✗	✗	✗	✗	✗

Note: “√” = Effect is present, “✗” = Effect is absent.

**Table 7 molecules-28-02217-t007:** Study design for behavioral and biochemical studies.

Groups	Treatment from Day 1–7
G-1 (Normal Control)	Normal saline 10 mL/Kg/p.o.
G-2 (Amnesic Control)	5% CMC 10 mL/Kg/p.o.
G-3 (Standard Control-A)	Piracetam 200 mg/Kg/p.o.
G-4 (Standard Control-B)	Piracetam 200 mg/Kg/p.o.
G-5 (Experimental Control-I)	P.aAF 3.5 mg/Kg/p.o.
G-6 (Experimental Control-II)	P.aAF 7 mg/Kg/p.o.
G-7 (Experimental Control-III)	P.aAF 7 mg/Kg/p.o.

Note: For preparation of oral doses, piracetam and scopolamine were dissolved in normal saline. However, P.aAF was suspended in 5% CMC. As per study design, P.aAF was administered daily for seven days to respective groups while scopolamine was only administered to G-2, G-4, G-5 and G-6 in a single oral dose on the 7th day of study. On the same day after 2 h of treatment, and on the next day, we performed the behavioral studies on mice, and finally the animals were dissected to perform biochemical evaluation on the 8th day.

**Table 8 molecules-28-02217-t008:** Study design for estimation of choline acetyltransferase (ChAT) levels in mice brains.

Groups	Treatment
G-A (Normal Control)	Normal saline 10 mL/Kg/p.o. for 7 days
G-B (Amnesic Control)	5% CMC 10 mL/Kg/p.o. for 6 days then Scopolamine 10 mg/Kg/p.o on 7th day.
G-C (Experimental Control-I)	P.aAF 7 mg/Kg/p.o. for 7 days consecutively
G-D (Experimental Control-II)	Scopolamine 10 mg/Kg/p.o on 1st day then P.aAF 7 mg/Kg/p.o. from day 2 to 7.
G-E (Experimental Control-III)	P.aAF 7 mg/Kg/p.o. for 6 days then Scopolamine 10 mg/Kg/p.o on 7th day.

Note: On 8th day, we finally dissected the animals and performed biochemical evaluation of ChAT levels on brain homogenates of the mice.

## Data Availability

All the data are shown in manuscript.

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
