# Peer review of "Biomolecular Screening of Pimpinella anisum L. for Antioxidant and Anticholinesterase Activity in Mice Brain"

_molecules, 2023, doi:10.3390/molecules28052217_

Round 1

Reviewer 1 Report

The authors conducted research entitled "Biomolecular Screening of Pimpinella anisum L. for Antioxidant & Anticholinesterase Activity in Mice Brain"

The manuscript seems interesting to me, however, at the same time it is limited in terms of robust results to be considered for publication in this important journal.

I recommend that the authors do a follow-up study to improve the quality of the manuscript. An in silico study, showing the molecular interactions of the major compounds with AChE, would be very important, are the authors willing to do this?

Other recommendations.

Review the state of the art of the introduction, and deepen the paragraphs, some references are very old and need to be more current.

Cite these references

https://doi.org/10.1016/j.tifs.2022.10.012

https://www.mdpi.com/1420-3049/27/20/7092

https://www.sciencedirect.com/science/article/pii/S025462992200059X

Materials and methods need to be organized.

Each step must be in a section or subsection.

Example: 1. Botanical material

1.2 Extraction

1.3 Fractionation, etc.

In the GC method, what is the concentration of the injected extract?

Was only qualitative analysis done? why not use CG-FID?

Improve the conclusion it is simple and without advancement.

Author Response

Thank you so much for reviewing our article and for worthy suggestions. We are highly obliged to the referee whose comments helped in improving this manuscript. We have revised the entire manuscript  for proper flow of the information. observations which are addressed in the point by point response are all incorporated into the revised file of the manuscript as per guidelines.

Following is the point to point response to the reviewer's comments: 

Reviewer's Comment: I recommend that the authors do a follow-up study to improve the quality of the manuscript. An in silico study, showing the molecular interactions of the major compounds with AChE, would be very important, are the authors willing to do this?

Authors Response: Thank you so much for the worthy suggestion. It is very valuable suggestion to incorporate an in-silico studies. But, unfortunately, at this stage, as we have complete the current project it will be very difficult to add on studies in the current study due to institutional allotted time frame limitations. We have noted this valuable point and we will surely apply this in our later on studies to improve the quality of research. Again bundle of thanks for the suggestion and guidelines. 

Other recommendations.

Reviewer's Comment: Review the state of the art of the introduction, and deepen the paragraphs, some references are very old and need to be more current.

Authors Response: Thank you so much for suggestion. We have added up materials in the introduction . 

Cite these references

https://doi.org/10.1016/j.tifs.2022.10.012

https://www.mdpi.com/1420-3049/27/20/7092

https://www.sciencedirect.com/science/article/pii/S025462992200059X

Authors Response: Thank you so much for suggestion. We have added up the suggested references. 

Reviewer's Comment: Materials and methods need to be organized.

Each step must be in a section or subsection.

Example: 1. Botanical material

1.2 Extraction

1.3 Fractionation, etc.

Authors Response: Thank you so much for suggestion. We made headings as per suggestion.

Reviewer's Comment: In the GC method, what is the concentration of the injected extract?

Authors Response: Thank you so much to mention the mistake. We have added concentration. It was 1 mg/mL

Reviewer's Comment: Was only qualitative analysis done? why not use CG-FID?

Authors Response: Thank you so much for the comment. In the current study, our focus was mainly on the qualitative analysis of the purified extract. That's why we used GC. It is a very good suggestion by the reviewer and in our next projects we will definitely focus on quantitative determination of organic contents of the extracts by using GC-FID.

Reviewer's Comment: Improve the conclusion it is simple and without advancement.

Authors Response: The reviewer's well said. We have made amendments in the conclusion. Now it is very clear and meaningful. 

Reviewer 2 Report

GC-MS is not an appropriate technique to analyze polar fractions, so it is recommended that authors should do LC-MS analysis to determine the secondary metabolites

Author Response

Thank you so much for reviewing our article and for worthy suggestion. We are highly obliged to the referee whose comments are much valuable for the quality of manuscript.

Following is the point to point response to the reviewer's comments: 

Reviewer's Comment: GC-MS is not an appropriate technique to analyze polar fractions, so it is recommended that authors should do LC-MS analysis to determine the secondary metabolites

Authors Response: Thank you so much for the suggestion and valuable comment. You right said GC-MS is not an appropriate technique to analyze polar fractions and is used mostly for analysis of essential oils. As mentioned in the materials and methods of the study that we prepared the different extracts in different solvents and then purified them by running in silica column. The aqueous fraction was run the column but the solvent system used as mobile phase was blend of polar and non-polar solvents as shown in Figure-1 of the manuscript. Then we analyzed that purified fraction in GC-MS by using methanol as solvent system. It is clear that Aqueous solvent is inappropriate to be used in GC-MS, so we further fractionated the aqueous fraction by column chromatography and then analyzed in GC. The reviewer's suggestion is very valuable that LC-MS can be used, but unfortunately due to unavailability of LC we used GC by involving methanol as solvent system.

The recommendations of two other referees are incorporated in the manuscript and revised file has been uploaded. 

Reviewer 3 Report

abstract: the sentences have no connection with each other, do not call the reader's attention,

Very short introduction, no mention of other works with plant extracts or isolated metabolites with the same activity

The authors concluded that the activity shown is due to the presence of oxadiazole compounds, but did not indicate any chemical structure.

How was the structural characterization of the active compounds (NMR???) performed? Molecular masses by interpretation of GC-MS library is not a structural identification

The very vague conclusion is based on what chemical structure?

The LD50 value is very high, can it be compared to what other substance or extract?

The extract is toxic

G1=G-I????, the table captions must contain the description of each group: G-1: Normal control etc

Very Confusing, GI, G-1, GA, multiple names for the same group 

The article is poorly written

If you had a GC, why didn't you isolate the substances and not evaluate them separately?

Conclusions should be based on the authors' results. 

Author Response

Thank you so much for reviewing our article and for worthy suggestions. We are highly obliged to the referee whose comments helped in improving this manuscript. We have revised the entire manuscript  for proper flow of the information. observations which are addressed in the point by point response are all incorporated into the revised file of the manuscript as per guidelines.

Reviewer's Comment: abstract: the sentences have no connection with each other, do not call the reader's attention,

Authors Response: Thank you so much for highlighting the issues. We have rewritten the abstract as per suggested comments. 

Reviewer's Comment: Very short introduction, no mention of other works with plant extracts or isolated metabolites with the same activity

Authors Response: Yes the reviewer is right and we have added the suggested materials in the introduction.

Reviewer's Comment: The authors concluded that the activity shown is due to the presence of oxadiazole compounds, but did not indicate any chemical structure.

Authors Response: Thank you so much for the valuable comment. We identified the compounds present in the extract by using GC-MS. The compounds were matched with GC-MS library are presented in Table 2 of the manuscript. Since, we do not have the facility of NMR that's why we did not claimed about the structure of the compounds.

Reviewer's Comment: How was the structural characterization of the active compounds (NMR???) performed? Molecular masses by interpretation of GC-MS library is not a structural identification

Authors Response: Thank you so much for your comment. We do not have NMR and were unable to identify the true chemical structure of the compounds. The compounds were identified on basis of information of GC-MS library data. The identification was based on the molecular masses and retention time as shown in table 2. 

Reviewer's Comment: The very vague conclusion is based on what chemical structure?

Authors Response: Agreed with reviewer, It has been modified as per recommendation.

Reviewer's Comment: The LD50 value is very high, can it be compared to what other substance or extract? The extract is toxic

Authors Response: We found LD-50 to evaluate the safety margin of the extract. It was calculate as 95 mg/Kg/Po. We did not compare it with any other substance. It can be done in further studies and detailed acute and chronic toxicity studies can be performed. 

In our point of view the extract is not toxic. The therapeutic dose at which maximum inhibition of AChE was observed was 7 mg/Kg/Po and LD 50 is calculated as 95. Hence the therapeutic window is quite broad. 

Reviewer's Comment: G1=G-I????, the table captions must contain the description of each group: G-1: Normal control etc

Very Confusing, GI, G-1, GA, multiple names for the same group

Authors Response: The reviewer right said. We appreciate the efforts. The highlighted mistakes have been removed and descriptions have been added in tables. 

Reviewer's Comment: The article is poorly written

Authors Response: Thanks to mention the quality of write up. Once the technical review will be completed then the manuscript will be finally checked by English language experts for the correction of wordings and grammar etc.

Reviewer's Comment: If you had a GC, why didn't you isolate the substances and not evaluate them separately?

Authors Response: Thank you for the comment. In previous studies we have found that the aqueous extract of P. anisum possesses strong potential to boost up memory. This qualitative study was designed to identify the constituents which are responsible for the nootropic activity. In future the study can be planned for the isolation of substances from the purified extracts along with quantitative analysis.

Reviewer's Comment: Conclusions should be based on the authors' results.

Agreed with reviewer, The conclusion has been modified as per recommendation.

Round 2

Reviewer 1 Report

Authors performed major revisions, manuscript may be processed for publication 

Reviewer 3 Report

The authors responded satisfactorily to the questions.